# Dietary Patterns are Associated with Leukocyte LINE-1 Methylation in Women: A Cross-Sectional Study in Southern Italy

**DOI:** 10.3390/nu11081843

**Published:** 2019-08-09

**Authors:** Martina Barchitta, Andrea Maugeri, Roberta Magnano San Lio, Giuliana Favara, Maria Clara La Rosa, Claudia La Mastra, Annalisa Quattrocchi, Antonella Agodi

**Affiliations:** Department of Medical and Surgical Sciences and Advanced Technologies “GF Ingrassia”, University of Catania, Via S. Sofia 87, 95123 Catania, Italy

**Keywords:** diet, epigenetics, DNA methylation, food intake

## Abstract

Bioactive food compounds have different effects on global DNA methylation, an epigenetic mechanism associated with chromosomal stability and genome function. Since the diet is characterized by a mixture of foods, we aimed to identify dietary patterns in women, and to evaluate their association with long interspersed nuclear elements (LINE-1) methylation, a surrogate marker of global DNA methylation. We conducted an observational cross-sectional study of 349 women from Southern Italy, with no history of severe diseases. Dietary patterns were derived by food frequency questionnaire and principal component analysis. LINE-1 methylation of leukocyte DNA was assessed by pyrosequencing. We observed that intake of wholemeal bread, cereals, fish, fruit, raw and cooked vegetables, legumes, soup, potatoes, fries, rice, and pizza positively correlated with LINE-1 methylation levels. By contrast, vegetable oil negatively correlated with LINE-1 methylation levels. Next, we demonstrated that adherence to a prudent dietary pattern—characterized by high intake of potatoes, cooked and raw vegetables, legumes, soup and fish—was positively associated with LINE-1 methylation. In particular, women in the 3rd tertile exhibited higher LINE-1 methylation level than those in the 1st tertile (median = 66.7 %5mC; IQR = 4.67 %5mC vs. median = 63.1 %5mC; IQR = 12.3 %5mC; *p* < 0.001). Linear regression confirmed that women in the 3rd tertile had higher LINE-1 methylation than those in the 1st tertile (β = 0.022; SE = 0.003; *p* < 0.001), after adjusting for age, educational level, employment status, smoking status, use of folic acid supplement, total energy intake and body mass index. By contrast, no differences in LINE-1 methylation across tertiles of adherence to the Western dietary pattern were evident. Interestingly, women who exclusively adhered to the prudent dietary pattern had a higher average LINE-1 methylation level than those who exclusively or preferably adhered to the Western dietary pattern (β = 0.030; SE = 0.004; *p* < 0.001; β = 0.023; SE = 0.004; *p* < 0.001; respectively), or those with no preference for a specific dietary pattern (β = 0.013; SE = 0.004; *p* = 0.002). Our study suggested a remarkable link between diet and DNA methylation; however, further mechanistic studies should be encouraged to understand the causal relationship between dietary intake and DNA methylation.

## 1. Introduction

Researchers, especially after the conclusion of the Human Genome Project, have raised questions about whether gene‒diet interaction may positively or negatively affect human health. With this question in mind, several research groups have begun to investigate the molecular mechanisms underpinning the effects of dietary interventions against aging and age-related diseases [1], but much effort is still needed to develop novel strategies for maintaining health and preventing disease [2]. Epigenetics—the study of molecular mechanisms that regulate gene expression without altering the DNA sequence [3]—has prompted much interest in elucidating the impact of diet and lifestyle choices on health. Among these mechanisms, DNA methylation has been extensively investigated in several diseases, including cardiovascular diseases, obesity, type-2 diabetes and cancer [4,5,6]. In mammals, DNA methylation is regulated by the activity of three DNA methyltransferases (DNMTs) and ten-eleven-translocation (TET) proteins [7]. DNA methylation often occurs within CpG islands, short sequences of 5–10 CpG dinucleotides per 100 bp located at gene promoters and regulatory regions [8]. In humans, nearly 80% CpG islands occur in repetitive sequences scattered throughout the genome, such as long interspersed nuclear elements (LINEs) and short interspersed nuclear elements (SINEs) [9]. LINE-1 elements—retrotransposons capable of independent and autonomous retro-transposition via RNA intermediate—comprise approximately 17% of the genome, with more than 500,000 copies. LINE-1 retro-transposition may lead to chromosomal instability, DNA rearrangement, and alteration of ectopic gene expression in tumour tissues [10]. In general, CpG islands located within LINE-1 sequences and their methylation levels correlate with the global genomic DNA methylation level [11]. For these reasons, LINE-1 methylation has been widely used as a surrogate marker of global genomic DNA methylation [12] in research on cancer, cardiovascular and neurodegenerative diseases [13,14,15,16].

Recent studies have proposed a remarkable link between epigenetic profile and environmental exposure, suggesting that nutrients, pollutants and other environmental factors can influence the turnover of epigenetic marks [17]. Thus, environmental and lifestyle factors can potentially modify DNA methylation, leading to genome reprogramming in exposed individuals and in future generations [18]. Notably, several nutrients (i.e., folate, polyphenols, selenium, retinoids, fatty acids, isothiocyanates and allyl compounds) and bioactive food compounds of fruit, vegetables and spices can influence epigenetic mechanisms by inhibiting the enzymes and substrates involved in DNA methylation or the histone modification processes [19,20,21,22,23]. Since diet is characterized by a mixture of foods and nutrients, research is moving towards the study of complex dietary patterns, rather than intake of a single food or nutrient. To the best of our knowledge, Zhang and colleagues were the first to report a positive association between a posteriori prudent dietary pattern—characterized by high intake of vegetables and fruits—and leukocyte LINE-1 methylation levels in a cancer-free population [24]. In line with this evidence, we recently demonstrated that adherence to the Mediterranean diet—evaluated by a priori Mediterranean diet score (MDS)—increased LINE-1 methylation level, counteracting the harmful effect of particulate matter exposure [25].

To determine whether dietary patterns may affect DNA methylation, we performed an observational cross-sectional study in women from Catania, Southern Italy. In this cohort, we first evaluated correlations between food intake and LINE-1 methylation. Next, we defined a posteriori dietary patterns that characterized the dietary habits of women, and investigated their association with leukocyte LINE-1 methylation level.

## 2. Materials and Methods

### 2.1. Study Design

Women with no history of severe diseases (i.e., cancer, CVD, diabetes, neurodegenerative and autoimmune diseases) were selected from those referred for routine physical examination to three clinical laboratories in Catania, Italy from 2010 to 2017. All women were fully informed of the purpose and invited to participate in the current observational study, which was carried out in accordance with the Declaration of Helsinki. The study protocol was approved by the ethics committees of the involved institutions. Information on sociodemographic and lifestyle characteristics were collected by trained epidemiologists using a structured questionnaire. We used data from non-pregnant women aged 12–87 years (median = 36 years), with complete assessment of social and behavioural characteristics and anthropometric measures. Education was categorized as low (including primary education or apprenticeship), medium (secondary education) or high (tertiary education) level. Women were also categorized according to their employment status as employed (including full-time or part-time employment) or unemployed (including retired). Smoking status was categorized as current, former or never. Body mass index (BMI) was calculated and categorized according to the World Health Organization (WHO) criteria [26].

### 2.2. Dietary Assessment

Dietary assessment was performed by a validated 95-item semi-quantitative FFQ, using the previous month as a reference period [21,25,27,28]. For each food item, women reported frequency of consumption (12 categories from “almost never” to “two or more times a day”) and portion size (small, medium or large), using a photograph atlas to minimize inaccuracies. Next, the daily intake was calculated by multiplying frequency by portion size, and adjusted for total energy intake using the residual method [29]. As fully described elsewhere [28,30,31,32], we selected 39 predefined food groups and performed a principal component analysis (PCA) with varimax rotation on energy-adjusted food group intakes. The selection of a posteriori dietary patterns to be retained was based on eigenvalues (>2.0), Scree plot inspection, and interpretability of components. Dietary patterns were labelled according to factor loadings with absolute value ≥0.3. The adherence to each dietary pattern was assessed using factor scores (i.e., the sum of the products between observed energy-adjusted food group intakes and their factor loadings), and categorized as low (1st tertile), medium (2nd tertile) or high (3rd tertile). Based on classification of both dietary patterns, we defined women as follows: those who were classified in opposite tertiles (1st vs. 3rd) were defined as exclusive adherents to a specific dietary pattern; those who were classified in adjacent tertiles (1st vs. 2nd or 2nd vs. 3rd) were defined as preferable adherents to a specific dietary pattern; those who were classified in the same tertile (1st vs. 1st, 2nd vs. 2nd or 3rd vs. 3rd) were defined as women with no preference for a specific dietary pattern.

### 2.3. LINE-1 Methylation Analysis

In recent decades, several methods have been proposed to determine levels of global genomic DNA methylation. These methods are based on high-performance liquid chromatography-ultraviolet, liquid chromatography coupled with tandem mass spectrometry, enzyme-linked immunosorbent assay, or pyrosequencing of bisulphite-converted DNA [33]. Alternatively, determining the methylation level of CpG sites located within LINE-1 sequences can be used as a surrogate marker of global DNA methylation level [11]. To allow comparison with our previous studies in this field of research, we evaluated the methylation level of three CpG sites within the LINE-1 sequence (GenBank Accession No. X58075) [21,25]. Each whole blood sample was centrifuged at 2500 rpm for 15 min, and the leukocyte fraction was immediately frozen at −20 °C until further analyses. From the leukocyte fraction, DNA was extracted using the QIAamp DNA Mini Kit (Qiagen, Milan, Italy) according to the manufacturer’s protocol. LINE-1 methylation level was measured by pyrosequencing-based methylation analysis of three CpG islands using the PyroMark Q24 instrument (Qiagen), as fully described elsewhere [21,34,35]. All assays were conducted in duplicate including positive (100% methylated DNA) and negative (0% methylated DNA) controls, while failed assays were repeated. In a previous study, we reported a 2.2% intra-observer coefficient of variability between the two replicates of LINE-1 methylation measurements (SD = 1.0%) [35]. For each CpG island, LNE-1 methylation levels were calculated as %5-mC over the total of cytosines. LINE-1 methylation level was reported for each CpG site and as their average.

### 2.4. Statistical Analysis

Statistical analyses were performed using the SPSS software (version 21.0, SPSS, Chicago, IL, USA). Continuous variables were tested for normality using the Kolmogorov‒Smirnov test, expressed as median and interquartile range (IQR), and compared using the Mann‒Whitney U test or Kruskal–Wallis test. Categorical variables were expressed as a frequency (percentage) and compared using the Chi-square test. Correlation analysis between food intakes and log-transformed LINE-1 methylation levels was performed using Spearman test followed by Bonferroni correction for 39 tests (*p* < 0.001 after Bonferroni correction). Next, we investigated the associations between dietary patterns and LINE-1 methylation levels using linear regression models on log-transformed data. In particular, we applied an unadjusted model (Model 1), followed by an adjusted model that included all the variables collected in our study (Model 2). All statistical tests were two-sided, and *p*-values < 0.05 were considered statistically significant, except for those reported after Bonferroni correction.

## 3. Results

### 3.1. Dietary Patterns of Study Population

The present analysis is based on 349 women, aged 12–87 years (median = 36 years), who were selected from those referred for routine physical examination to three clinical laboratories in Catania, Italy. In this cohort, we derived two major dietary patterns with eigenvalues ≥2.0, which explained 17.2% of the total variance among 39 predefined food groups. Figure 1 shows factor loadings, which can be viewed as the correlation between each food group and dietary pattern. The first dietary pattern—named “prudent”—was characterized by high intake of potatoes, cooked and raw vegetables, legumes, soup and fish. As shown in Table 1, age increased from the bottom to the top tertile of adherence to the “prudent” dietary pattern. Likewise, women in the top tertile had a higher total energy intake than those in the bottom tertile. By contrast, women in the top tertile were less likely to take folic acid supplements than those in the bottom tertile. We also observed a slight but significant difference in the distribution of educational level across tertiles.

The second dietary pattern—named “Western”—was characterized by a high intake of canned fish, vegetable oil, processed meat, salty snacks, alcoholic drinks and dipping sauces, and low intake of fruits. Contrary to the “prudent” dietary pattern, we observed that age decreased from the bottom to the top tertile of adherence to the “Western” dietary, while no significant differences were evident for other socio-demographic variables. Interestingly, the total energy intake showed a U-shaped distribution, with lower levels in the second tertile than in the other tertiles. Moreover, being in the top tertile of adherence to the “Western” dietary pattern was associated with a higher prevalence of never smokers and obesity.

### 3.2. Correlations between Food Intake and LINE-1 Methylation

Overall, LINE-1 methylation levels exhibited a skewed distribution with the median level of 65.0 %5mC (IQR = 7.5 %5mC). As shown in Figure 2, we first reported significant correlations between intake of 39 food groups and LINE-1 methylation after Bonferroni correction. Notably, we found that wholemeal bread, cereals, fish, fruit, raw and cooked vegetables, legumes, soup, potatoes, fries, rice and pizza were positively correlated with average LINE-1 methylation. By contrast, vegetable oil was negatively correlated with average LINE-1 methylation levels. Figure 2 also displays similar correlations between food intake and LINE-1 methylation at CpG sites 1, 2 and 3.

### 3.3. Association between Dietary Patterns and LINE-1 Methylation

Figure 3 showed differences in LINE-1 methylation levels according to adherence to dietary patterns. With respect to the “prudent” dietary pattern, LINE-1 methylation levels at the three CpG sites increased from the bottom to the top tertile (Figure 3a). In particular, women in the top tertile of adherence reported higher LINE-1 methylation levels in CpG site 1 (median = 81.0 %5mC; IQR = 3.0 %5mC), CpG site 2 (median = 54.0 %5mC; IQR = 6.0 %5mC) and CpG site 3 (median = 64.0 %5mC; IQR = 6.0 %5mC) than those in the bottom tertile (median = 80.0 %5mC; IQR = 4.0 %5mC; median = 51.0 %5mC; IQR = 21.0 %5mC; median = 59.5 %5mC; IQR = 16.5 %5mC, respectively). In line with these findings, women in the top tertile of adherence to the “prudent” dietary pattern exhibited a higher average LINE-1 methylation level (median = 66.7 %5mC; IQR = 4.67 %5mC) than those in the bottom tertile (median = 63.1 %5mC; IQR = 12.3 %5mC) (*p* < 0.001). By contrast, no differences in LINE-1 methylation levels across tertiles of adherence to the “Western” dietary pattern were evident (Figure 3b).

Linear regression analysis on log-transformed data confirmed the increasing trend of LINE-1 methylation across tertiles of adherence to the “prudent” dietary pattern in the unadjusted model (Model 1), and we further adjusted for age, educational level, employment status, smoking status, use of folic acid supplement, total energy intake and BMI (Model 2) (Table 2). In particular, being in the top tertile of adherence to the “prudent” dietary pattern was associated with higher LINE-1 methylation levels at CpG site 1 (β = 0.009; SE = 0.003; *p* = 0.001), CpG site 2 (β = 0.030; SE = 0.005; *p* < 0.001) and CpG site 3 (β = 0.034; SE = 0.003; *p* < 0.001), after adjusting for covariates. In line with these findings, we demonstrated that women in the top tertile had a higher average LINE-1 methylation levels than those in the bottom tertile (β = 0.022; SE = 0.003; *p* < 0.001). By contrast, no association between “Western” dietary pattern and LINE-1 methylation was evident, in the unadjusted model (Model 1) and further adjusting for covariates (Model 2) (Table 3). Among the covariates included in the regression model, only increasing age (*p* < 0.001) and total energy intake (*p* < 0.001) were positively associated with LINE-1 methylation level.

Finally, we compared LINE-1 methylation levels between women who exclusively or preferably adhered to a specific dietary pattern and those with no preference (Figure 4). We observed a significant increasing trend of LINE-1 methylation level from women who exclusively adhered to the Western dietary pattern to those who adhered to the prudent dietary pattern (*p* < 0.001 for CpG1, CpG3, and average LINE-1 methylation; *p* = 0.009 for CpG2). Interestingly, these findings were confirmed by linear regression analysis. Indeed, women who exclusively adhered to the prudent dietary pattern had a higher average LINE-1 methylation level than those who exclusively adhered to the Western dietary pattern in the unadjusted (β = 0.024; SE = 0.008; *p* = 0.004) and adjusted models (β = 0.030; SE = 0.004; *p* < 0.001). In the adjusted model, we also demonstrated that women who exclusively adhered to the prudent dietary pattern had a higher average LINE-1 methylation level than those who preferably adhered to the Western dietary pattern (β = 0.023; SE = 0.004; *p* < 0.001), or those with no preference for a specific dietary pattern (β = 0.013; SE = 0.004; *p* = 0.002).

## 4. Discussion

The present study aimed to investigate the relationship of food intakes and dietary patterns with LINE-1 methylation in healthy women from Southern Italy. Methylation of LINE-1 sequences has been widely considered a surrogate marker of global genomic DNA methylation [12] in research into cancer, cardiovascular and neurodegenerative diseases [13,14,15]. In particular, hypomethylation of these sequences has been associated with chromosomal instability and aberrant genome function, due to LINE-1 retro-transposition and insertion throughout the genome. [36,37]. In general, it has been estimated that LINE-1 retro-transposition accounts for at least one in every 50 humans within a parental germ cell or during early foetal development [38]. Thus, LINE-1 hypomethylation in the parental germline, along with altered miRNA expression, might also significantly affect genome stability during the foetal development [39,40].

In our study, we first found that intake of “healthy” foods—such as wholemeal bread, cereals, fish, fruit, raw and cooked vegetables, legumes, and soup—was positively correlated with LINE-1 methylation. These findings partially confirmed previous studies reporting that higher intake of vegetables and/or fruits decreased the risk of LINE-1 hypomethylation [21,24]. The biological explanation of this relationship could be attributed to the wide variety of nutrients and bioactive compounds provided by fruits and vegetables—including phytochemicals (phenolics, flavonoids, and carotenoids), vitamins (vitamin C, folate, and pro-vitamin A), minerals (potassium, calcium, and magnesium), and fibre—which in turn modulate multiple pathways associated with epigenetic mechanisms [41,42]. By contrast, we observed that intake of vegetable oil seemed to be negatively correlated with LINE-1 methylation. The study by de la Rocha and colleagues partially supported these results, pointing out the association between whole peripheral blood fatty acids and DNA methylation, measured as total level of 5-methyldeoxycytosine [43]. However, further research is needed to better elucidate the effect of dietary fat intake on LINE-1 methylation.

Previous findings focused on the effect of specific foods on the DNA methylation process, but there is currently growing interest in determining how dietary patterns may affect global and local DNA methylation in humans. To solve this question, Zhang and colleagues examined the association between dietary patterns and leukocyte LINE-1 methylation in 149 individuals with no history of cancer [24]. First, they applied PCA on 13 food groups to derive two dietary patterns: a “prudent” one, characterized by a high intake of vegetables and fruits; and the “Western” one, characterized by a high intake of energy-dense foods such as grains, meats, fries, oil, and dairy. While only the intake of dark green vegetables seemed to be significantly associated with LINE-1 methylation among specific food groups, the analysis of dietary patterns revealed a positive association between the prudent dietary pattern and LINE-1 methylation in a dose‒response manner [24].

In our cohort of 349 women with no history of severe diseases (i.e., cancer, cardiovascular disease, diabetes, neurodegenerative and autoimmune diseases), we identified two dietary patterns by applying PCA on 39 food groups. Similar to Zhang and colleagues, we derived the “prudent” dietary pattern—characterized by high intake of potatoes, cooked and raw vegetables, legumes, soup and fish—and the “Western” dietary pattern—characterized by high intake of canned fish, vegetable oil, processed meat, salty snacks, alcoholic drinks and dipping sauces, and low intake of fruits. Interestingly, LINE-1 methylation levels at the three CpG sites and their average increased from the bottom to the top tertile of adherence to the “prudent” dietary pattern. After adjusting for age, educational level, use of folic acid supplement and total energy intake, we confirmed that high adherence to the “prudent” dietary pattern was associated with higher LINE-1 methylation levels at CpG site 1 and CpG site 3. In line with these findings, women with high adherence to the “prudent” dietary pattern also exhibited higher average LINE-1 methylation levels than those with low adherence. By contrast, no association between the “Western” dietary pattern and LINE-1 methylation was evident, also adjusting for covariates. We also observed a significant increasing trend of LINE-1 methylation level from women who exclusively adhered to the Western dietary pattern to those who adhered to the prudent dietary pattern. Interestingly, we found that women who exclusively adhered to the prudent dietary pattern had higher average LINE-1 methylation level than those who exclusively or preferably adhered to the Western dietary pattern, or those with no preference for a specific dietary pattern.

This study has several strengths. Compared to previous studies, our results were obtained from a larger cohort of individuals with no history of severe diseases. Moreover, data collection was performed through standard and validated tools, which enabled us to apply PCA on 39 food groups. With respect to DNA methylation analysis, we applied the pyrosequencing of bisulphite-treated DNA, which is a replicable methodology and the “gold standard” to evaluate LINE-1 methylation levels [44,45]. Finally, our results were robust, as they have been confirmed after adjusting for several socio-demographic and behavioural covariates.

However, this study also has some limitations. Its observational, cross-sectional design did not allow us to assess causality between adherence to dietary patterns and LINE-1 methylation. Previous in vitro studies described the potential molecular mechanisms underpinning this relationship, and those affecting genome and chromosome stability [38]. However, more efforts are needed to understand the causal link between diet, LINE-1 methylation and genome/chromosomal instability in humans. Dietary data collection was performed using FFQs, which did not preclude measurement errors and inaccuracy. However, weighted records and 24-h recalls—among the most commonly used tools for dietary assessment—are also prone to a degree of misreporting [46]. Although great strides have been made in the accuracy of dietary intake assessment methods, the development of open-ended methods with various innovative technologies should be encouraged [46]. Nowadays, since novel methods are more expensive than FFQs, with unresolved intrinsic problems also related to self-reporting, FFQs remain widely used as a primary dietary assessment tool in epidemiological studies [46]. Despite the limitations, the FFQ used in the present study has been previously developed and validated among a similar cohort of women from Southern Italy [47]. In addition, our PCA-derived dietary patterns were consistent with previous studies from different European countries [28,48,49,50,51]. Moreover, we performed LINE-1 methylation analysis on leukocyte DNA, which included several cell type subsets. However, previous studies reported small differences in LINE-1 methylation levels according to the blood cell composition [52,53,54,55,56]. Others also described LINE-1 methylation differences across different CpG sites and tissues [33]. Overall, these potential differences discourage the comparison of our results with those reported by previous researchers [34], and we also cannot completely avoid bias due to changes in the density of leukocyte populations. Finally, we cannot completely exclude the effect of unmeasured residual factors, such as ethnicity, drinking, physical activity and environmental exposure. Moreover, the association between diet and LINE-1 methylation may also be affected by genetic factors (e.g., polymorphisms in the *MTHFR* gene), which in turn may interact with the methylation process [57].

## 5. Conclusions

In conclusion, we described a positive correlation between the intake of “healthy” foods and LINE-1 methylation levels, which in turn was negatively associated with chromosomal instability and aberrant genome function. Notably, we demonstrated that women with high adherence to the “prudent” dietary pattern—rich in potatoes, cooked and raw vegetables, legumes, soup and fish—had an increased LINE-1 methylation level compared to those who adhered to a Western diet. Although our study suggested a potential association between LINE-1 methylation and adherence to a healthy diet, further outcome-driven research and mechanistic studies should be encouraged to evaluate their causal relationship.

## Figures and Tables

**Figure 1 nutrients-11-01843-f001:**
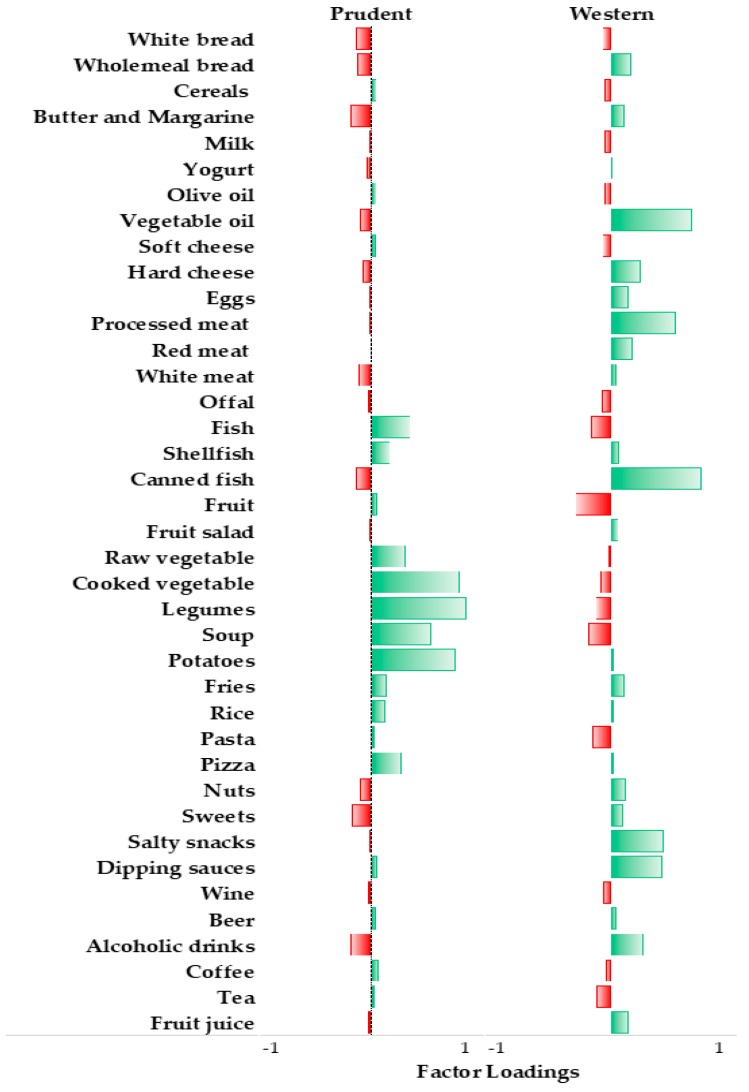
Bar graph of factor loadings characterizing dietary patterns.

**Figure 2 nutrients-11-01843-f002:**
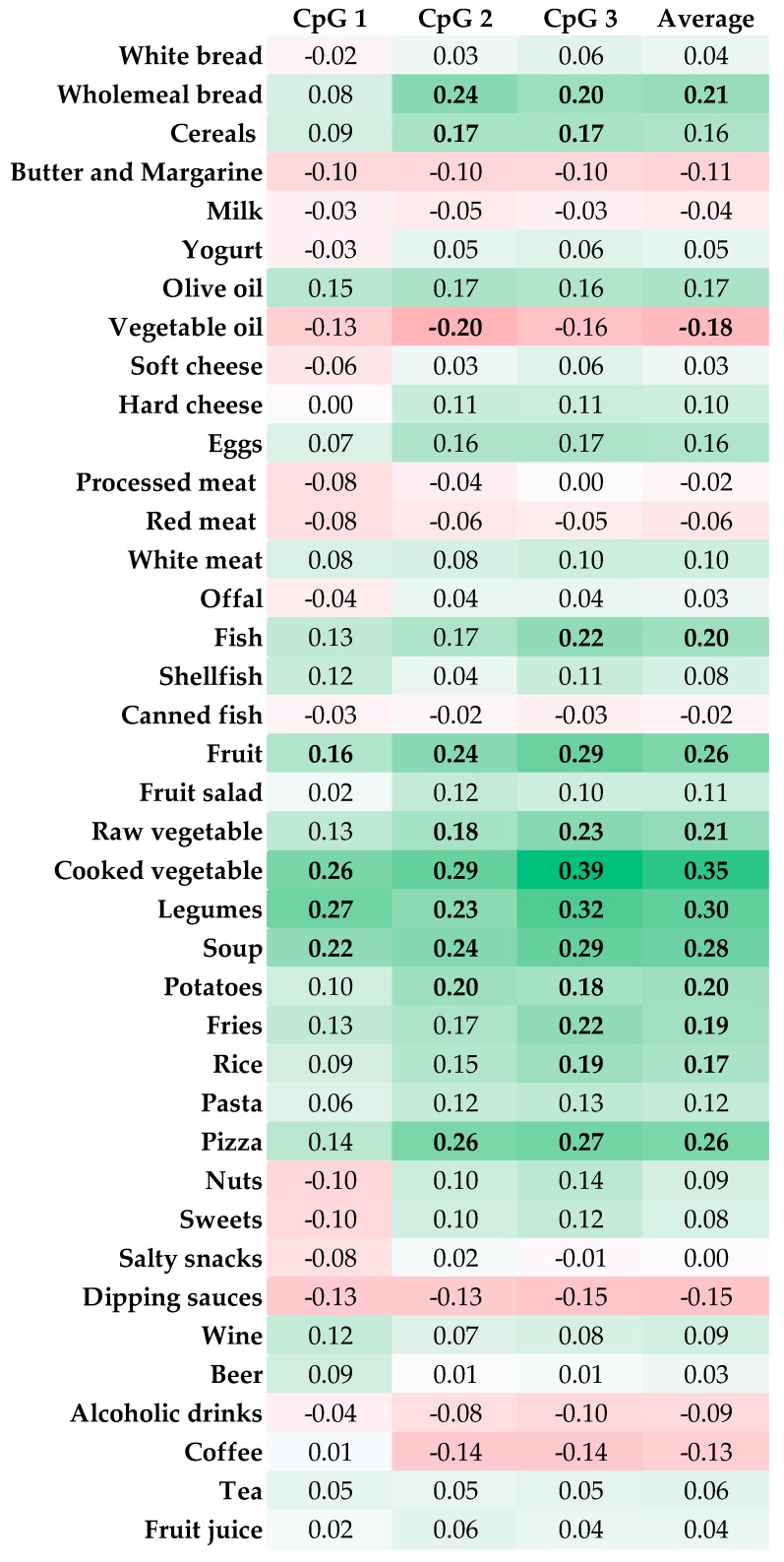
Correlation matrix between food intake and log-transformed LINE-1 methylation level. Results are reported as Spearman’s correlation coefficient and those with Bonferroni-corrected *p*-value < 0.001 are indicated in bold font.

**Figure 3 nutrients-11-01843-f003:**
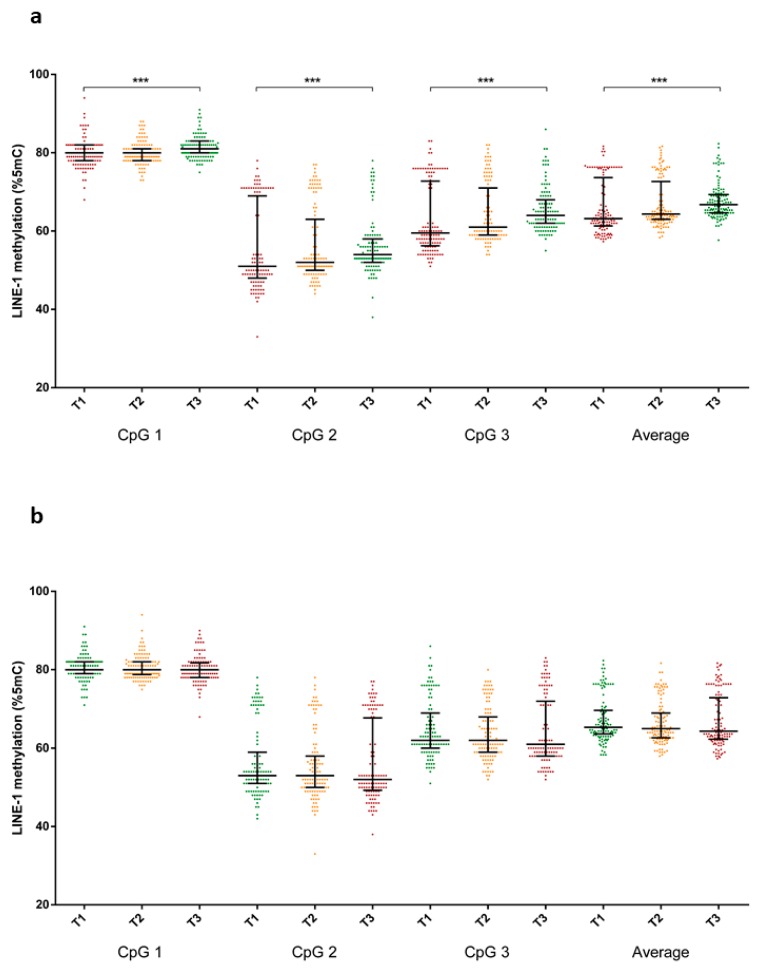
Comparison of LINE-1 methylation level across tertiles of adherence to (**a**) the prudent and (**b**) Western dietary patterns. *** *p* < 0.001 based on the Kruskal–Wallis test.

**Figure 4 nutrients-11-01843-f004:**
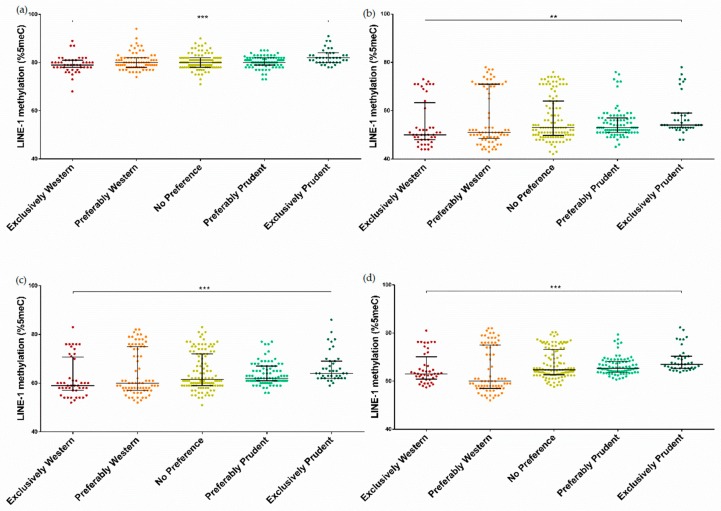
Comparison of LINE-1 methylation level across categories of adherence to dietary patterns for CpG site 1 (**a**), CpG site 2 (**b**), CpG site 3 (**c**) and their average (**d**). ** *p* < 0.01 and *** *p* < 0.001, based on the Kruskal–Wallis test.

**Table 1 nutrients-11-01843-t001:** Characteristics of study population by adherence to dietary patterns.

Characteristics	Prudent				Western			
	1st Tertile	2nd Tertile	3rd Tertile	*p*-Value	1st Tertile	2nd Tertile	3rd Tertile	*p*-Value
Age, years	32.0 (19.0)	35.0 (19.0)	46.0 (34.0)	<0.001	41.0 (34.0)	40.0 (24.0)	30.0 (17.0)	<0.001
Educational level								
Low	34.5%	18.8%	23.3%	0.049	24.1%	23.1%	29.3%	0.832
Medium	37.9%	53.8%	50.9%		47.4%	49.6%	45.7%	
High	27.6%	27.4%	25.9%		28.4%	27.4%	25.0%	
Employment status (% unemployed)	62.1%	47.0%	53.4%	0.069	53.4%	55.6%	53.4%	0.933
Smoking status								
Never smokers	67.8%	69.2%	62.6%	0.757	63.5%	65.5%	70.7%	0.011
Former smokers	11.3%	13.7%	14.8%		14.8%	19.8%	5.2%	
Current smokers	20.9%	17.1%	22.6%		21.7%	14.7%	24.1%	
Use of folic acid supplement (% users)	19.0%	18.8%	7.8%	0.024	19.8%	16.2%	9.5%	0.083
Total energy intake, kcal	1693.0 (3581.0)	1940.0 (614.0)	2142.0 (563.0)	<0.001	2028.0 (682.0)	1781.0 (2610.0)	2065.0 (744.0)	0.001
Body mass index, kg/m^2^	23.3 (6.8)	23.9 (5.8)	24.1 (23.0)	0.971	23.0 (4.8)	25.0 (5.7)	23.0 (8.1)	0.067
Body mass index categories								
Underweight	3.5%	6.9%	5.2%	0.403	3.5%	2.6%	9.6%	0.002
Normal weight	60.9%	50.9%	53.0%		65.2%	47.9%	51.8%	
Overweight	19.1%	28.4%	29.6%		21.7%	35.9%	19.3%	
Obese	16.5%	13.8%	12.2%		9.6%	13.7%	19.3%	

Results are reported as median (interquartile range), or percentage. Statistical analyses were performed using Chi-square test for bivariate or categorical variable, and Kruskal–Wallis test for continuous variables.

**Table 2 nutrients-11-01843-t002:** Linear regression analysis of the association between adherence to the prudent dietary pattern and LINE-1 methylation level.

Regression Model	LINE-1 Methylation	1st Tertile	2nd Tertile		3rd Tertile		*p*-Trend
			β (SE)	*p*-Value	β (SE)	*p*-Value	
Model 1	CpG site 1	*Ref.*	0.001 (0.002)	0.599	0.008 (0.002)	<0.001	<0.001
	CpG site 2	*Ref.*	0.009 (0.010)	0.348	0.011 (0.009)	0.234	0.233
	CpG site 3	*Ref.*	0.011 (0.007)	0.120	0.019 (0.006)	0.003	0.003
	Average	*Ref.*	0.006 (0.005)	0.263	0.012 (0.005)	0.017	0.017
Model 2	CpG site 1	*Ref.*	0.001 (0.002)	0.990	0.009 (0.003)	0.001	<0.001
	CpG site 2	*Ref.*	0.015 (0.004)	0.001	0.030 (0.005)	<0.001	<0.001
	CpG site 3	*Ref.*	0.016 (0.003)	<0.001	0.034 (0.003)	<0.001	<0.001
	Average	*Ref.*	0.009 (0.002)	<0.001	0.022 (0.003)	<0.001	<0.001

Statistical analysis was performed using unadjusted linear regression (Model 1) and further adjusting for age, educational level, employment status, smoking status, use of folic acid supplement, total energy intake and BMI (Model 2).

**Table 3 nutrients-11-01843-t003:** Linear regression analysis of the association between adherence to the Western dietary pattern and LINE-1 methylation level.

Regression Model	LINE-1 Methylation	1st Tertile	2nd Tertile		3rd Tertile		*p*-Trend
			β (SE)	*p*-Value	β (SE)	*p*-Value	
Model 1	CpG site 1	*Ref.*	0.001 (0.002)	0.828	−0.003 (0.002)	0.276	0.262
	CpG site 2	*Ref.*	−0.009 (0.008)	0.310	−0.002 (0.009)	0.838	0.835
	CpG site 3	*Ref.*	−0.008 (0.006)	0.202	−0.002 (0.007)	0.753	0.743
	Average	*Ref.*	−0.005 (0.005)	0.316	−0.002 (0.005)	0.702	0.690
Model 2	CpG site 1	*Ref.*	0.002 (0.002)	0.523	−0.001 (0.003)	0.676	0.760
	CpG site 2	*Ref.*	0.005 (0.005)	0.282	−0.003 (0.005)	0.549	0.837
	CpG site 3	*Ref.*	0.007 (0.004)	0.067	−0.001 (0.004)	0.834	0.705
	Average	*Ref.*	0.004 (0.003)	0.101	−0.001 (0.003)	0.647	0.986

Statistical analysis was performed using unadjusted linear regression (Model 1) and further adjusting for age, educational level, employment status, smoking status, use of folic acid supplement, total energy intake and BMI (Model 2).

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
