# Peer review of "Dietary Patterns are Associated with Leukocyte LINE-1 Methylation in Women: A Cross-Sectional Study in Southern Italy"

_nutrients, 2019, doi:10.3390/nu11081843_

Round 1

Reviewer 1 Report

The authors have revised the manuscript according to my suggestions. There is one remaining issue to consider:

-In the correlation analysis between foods and LINe-1 methylation, was LINE-1 methylation log-transformed? As this has been done later to normalise the distribution, it should be done here too.

Author Response

Dear Editor,

Thank you very much for considering our manuscript and for suggestions of independent Reviewers. We submit to your attention a revised version of the manuscript in which we have considered all comments. The following List of change and answers to comments of Reviewers addresses all changes made in the manuscript (blue font).We also edited some parts of our manuscript according to the Editor’s requests.  

Reviewer 1

The authors have revised the manuscript according to my suggestions. There is one remaining issue to consider:

-In the correlation analysis between foods and LINe-1 methylation, was LINE-1 methylation log-transformed? As this has been done later to normalise the distribution, it should be done here too.

We are very grateful with Reviewer 1 for his/her comments that helped us in improving our manuscript. As requested, we addressed the remaining issue the on log-transformation of LINE-1 methylation levels prior to test correlation. Accordingly, we have modified the statistical analysis section and figure 2.

Reviewer 2 Report

The authors have explored if global DNA methylation levels can be affected by diet. They have performed their analysis on whole blood samples from 349 healthy women from Southern Italy. Global DNA methylation changes were assessed through the measurement of LINE-1 methylation levels, via pyrosequencing of 3 CpGs, a commonly used indirect approach to assess global DNA methylation levels.

Main comments

I do not have any specific comments on methods and statistical analyses, discussion - the authors have described very well the pluses and limitations of their study. However, I suggest the following minor changes:

1.       Results – Figure 3 ‘Association between dietary patterns and LINE-1 methylation’.

The authors focus on the statistical differences between tertiles within one dietary pattern. While this is fine, because it gives a notion of a quantitative dose-dependent effect from the extent of adherence to a diet (i.e. for the ‘Prudent’, as no such effect is seen for the ‘Western’ diet), and hence this suggests there could be a real correlation or causality between the diet and LINE-1 methylation levels, I am sure many readers like me would want to know what differences (if any) there are between the ‘Prudent’ and ‘Western’ diets’ methylation levels compared to each other directly. The way figure 3 is presented, it is very difficult for the reader to assess visually what the methylation differences are for each CpG between the two diets and compare them. Given that the levels seem overall very similar, it is important to assess this with proper statistical methods, and not leave the reader to guess and struggle to compare visually.

I would suggest, if possible, to restructure Fig. 3, by presenting the same data in 4 plots, rather than 2, structured like this: each of four plots represents a single CpG, or the Average value, and shows T1T2T3 for both diets on the X axis next to each other. Thus, the authors can still do their stats for the tertiles for each CpG within a dietary pattern, as currently done, but can also assess statistically any differences in CpG or Average methylation between the different diets. For the latter T3s could be most interesting – is the increase in Prudent T3 methylation level also significantly higher than all or any of the T1T2T3 levels for the Western diet for the same CpG/Average, or not? For me this is the main question for this study, and currently it is not answered.

2.       Conclusions – the authors use LINE-1 methylation as a surrogate for global DNA methylation levels. And yet, their main conclusion is not ‘strict adherence to prudent diet increases global DNA methylation levels’ but instead, ‘LINE-1 methylation is a molecular mechanism underpinning the protective effect of prudent dietary pattern in healthy women.’ I think the latter conclusion is too far stretched, without any mechanistic evidence. See also specific comment for Lines 318-319.

Specific comments:

Line 11 – Abstract – the first sentence is very confusing – can the authors please rephrase. “Bioactive food compounds can affect global DNA methylation (here it does not become clear how they can affect – by changing levels, patterns, distribution etc?), which in turn is negatively associated with chromosomal instability and genome function.” > high global DNA methylation is associated with genome stability, so the second part of the sentence suggests that the authors mean that ‘Bioactive food compounds can increase global DNA methylation levels’ – if so, please add ‘levels’ after DNA methylation, but also specify what you mean by ‘can affect’ (i.e. increase?).

Line 20 – I would add ‘levels’ after LINE-1 methylation in both sentences. Methylation has levels, distribution, patterns, profile, etc so it is important to describe in what way methylation is affected, i.e. in this case - its levels.

Line 47 – “This process almost exclusively occurs within CpG islands,…” > it is not clear what ‘this process’ is following from the previous sentence. If the authors mean that DNA methylation occurs exclusively in CpG islands, I would disagree, because CGIs are known as mostly unmethylated, while the rest of the CpGs in the genome are highly methylated – meaning that DNA methylation occurs largely outside of CGIs. Please rephrase, the sentence is currently confusing.

Line 59 – the authors have written ‘a remarkable link between epigenetic and environmental exposure’ – there is no such thing as ‘epigenetic exposure’, so the authors need to specify “epigenetic what? and environmental exposure”; suggestions: ‘the epigenetic landscape’, ‘the epigenetic profile’, etc.

Line 70 – written ‘leukocyte LINE-1 methylation’, again the noun that characterises the methylation is omitted, please add ‘levels’ after methylation to specify what the positive association is with – assuming LINE-1 methylation levels is what the authors are talking about here.

Lines 239-241 – “it has been estimated that LINE-1 insertions account for at least 1 in every 50 humans within a parental germ cell or during early fetal development.“ > the sentence misses something, it does not become clear what LINE-1 insertions account for – mutations or what? Please revise the sentence and change so it makes sense.

Lines 318-319 – the authors have stated “Although our study suggested LINE-1 methylation as a molecular mechanism underpinning the protective effect of healthy diet” > this is quite a strong statement. There are around 30 million CpGs in the human genome, a large proportion of them are methylated. The genomic stability depends on maintaining the methylation status of many of those CpGs, whether in repeats, promoters, gene bodies or elsewhere. Measuring 3 CpGs in one of the common repeats (of 500,000 copies), which could indeed cause problems when hypomethylated, could definitely suggest one of the mechanisms, but it is a very strong statement to suggest this is the “molecular mechanism underpinning the protective effect of healthy diet”. The authors could make a point by providing some background on numbers (if those 3 CpGs actually occur 500,000 times each and make 1,5 million CpGs in the genome in total this is impressive), but nevertheless, I would tone this down here and also in the abstract. The prudent diet supposes a higher intake of folic acid, which helps to maintain higher global DNA methylation levels (which otherwise tend to erode with aging) – but this effect is most likely seen throughout the whole methylome, not only in LINE-1 (which is also why LINE-1 is used as an indirect measure for the global methylation levels). 

Author Response

Dear Editor,

Thank you very much for considering our manuscript and for suggestions of independent Reviewers. We submit to your attention a revised version of the manuscript in which we have considered all comments. The following List of change and answers to comments of Reviewers addresses all changes made in the manuscript (blue font).We also edited some parts of our manuscript according to the Editor’s requests.  

Reviewer 2

The authors have explored if global DNA methylation levels can be affected by diet. They have performed their analysis on whole blood samples from 349 healthy women from Southern Italy. Global DNA methylation changes were assessed through the measurement of LINE-1 methylation levels, via pyrosequencing of 3 CpGs, a commonly used indirect approach to assess global DNA methylation levels.

Main comments

I do not have any specific comments on methods and statistical analyses, discussion - the authors have described very well the pluses and limitations of their study. However, I suggest the following minor changes:

We are very grateful with Reviewer 2 for his/her comments that helped us in improving our manuscript.

1.       Results – Figure 3 ‘Association between dietary patterns and LINE-1 methylation’.

The authors focus on the statistical differences between tertiles within one dietary pattern. While this is fine, because it gives a notion of a quantitative dose-dependent effect from the extent of adherence to a diet (i.e. for the ‘Prudent’, as no such effect is seen for the ‘Western’ diet), and hence this suggests there could be a real correlation or causality between the diet and LINE-1 methylation levels, I am sure many readers like me would want to know what differences (if any) there are between the ‘Prudent’ and ‘Western’ diets’ methylation levels compared to each other directly. The way figure 3 is presented, it is very difficult for the reader to assess visually what the methylation differences are for each CpG between the two diets and compare them. Given that the levels seem overall very similar, it is important to assess this with proper statistical methods, and not leave the reader to guess and struggle to compare visually.

I would suggest, if possible, to restructure Fig. 3, by presenting the same data in 4 plots, rather than 2, structured like this: each of four plots represents a single CpG, or the Average value, and shows T1T2T3 for both diets on the X axis next to each other. Thus, the authors can still do their stats for the tertiles for each CpG within a dietary pattern, as currently done, but can also assess statistically any differences in CpG or Average methylation between the different diets. For the latter T3s could be most interesting – is the increase in Prudent T3 methylation level also significantly higher than all or any of the T1T2T3 levels for the Western diet for the same CpG/Average, or not? For me this is the main question for this study, and currently it is not answered.

We strongly agree with this comment and suggestion that may really improve our manuscript. Thus, in the revised version of our manuscript, we also classified women according to their adherence to dietary patterns as follows:

·       exclusively adherents to a specific dietary pattern (those who were classified in opposite tertiles)

·       preferably adherents to a specific dietary pattern (those who were classified in adjacent tertiles)

·       no preference for a specific dietary pattern (those who were classified in the same tertile)

Next, we compared LINE-1 methylation levels from women who exclusively adhered to the western dietary pattern to those who exclusively adhered to the prudent dietary pattern (please consider Figure 4). Finally, we compared average LINE-1 methylation levels of women who exclusively adhered to the prudent dietary pattern with those who exclusively or preferably adhered to the western dietary pattern, and those with no preference. We think that these additional analyses make our manuscript more interesting, and we really want to thank Reviewer 2. Indeed, as described in the result section, we observed a significant increasing trend of LINE-1 methylation level from women who exclusively adhered to the western dietary pattern to those who adhered to the prudent dietary pattern (p<0.001 for CpG1, CpG3, and average LINE-1 methylation; p=0.009 for CpG2). Moreover, women who exclusively adhered to the prudent dietary pattern had higher average LINE-1 methylation level than those who exclusively adhered to the western dietary pattern in the unadjusted (β= 0.024; SE= 0.008; p=0.004) and adjusted models (β= 0.030; SE= 0.004; p< 0.001). In the adjusted model, we also demonstrated that women who exclusively adhered to the prudent dietary pattern had higher average LINE-1 methylation level than those who preferably adhered to the western dietary pattern (β= 0.023; SE= 0.004; p< 0.001), or those with no preference for a specific dietary pattern (β= 0.013; SE= 0.004; p=0.002).

2.       Conclusions – the authors use LINE-1 methylation as a surrogate for global DNA methylation levels. And yet, their main conclusion is not ‘strict adherence to prudent diet increases global DNA methylation levels’ but instead, ‘LINE-1 methylation is a molecular mechanism underpinning the protective effect of prudent dietary pattern in healthy women.’ I think the latter conclusion is too far stretched, without any mechanistic evidence. See also specific comment for Lines 318-319.

We apologize if our conclusion was overstated and we made some changes accordingly.

Specific comments:

Line 11 – Abstract – the first sentence is very confusing – can the authors please rephrase. “Bioactive food compounds can affect global DNA methylation (here it does not become clear how they can affect – by changing levels, patterns, distribution etc?), which in turn is negatively associated with chromosomal instability and genome function.” > high global DNA methylation is associated with genome stability, so the second part of the sentence suggests that the authors mean that ‘Bioactive food compounds can increase global DNA methylation levels’ – if so, please add ‘levels’ after DNA methylation, but also specify what you mean by ‘can affect’ (i.e. increase?).

To our knowledge, bioactive food compounds may have different effects on DNA methylation, and this depend on the type of compound, DNA sequence under investigation, and additional factors. As suggested, we rephrased the first sentence of the abstract and added “levels” to DNA methylation. 

Line 20 – I would add ‘levels’ after LINE-1 methylation in both sentences. Methylation has levels, distribution, patterns, profile, etc so it is important to describe in what way methylation is affected, i.e. in this case - its levels.

As suggested, we added “levels” after LINE-1 methylation. 

Line 47 – “This process almost exclusively occurs within CpG islands,…” > it is not clear what ‘this process’ is following from the previous sentence. If the authors mean that DNA methylation occurs exclusively in CpG islands, I would disagree, because CGIs are known as mostly unmethylated, while the rest of the CpGs in the genome are highly methylated – meaning that DNA methylation occurs largely outside of CGIs. Please rephrase, the sentence is currently confusing.

According to Reviewer 2 suggestion, we rephrased this sentence indicating that DNA methylation often occurs in CpG islands.

Line 59 – the authors have written ‘a remarkable link between epigenetic and environmental exposure’ – there is no such thing as ‘epigenetic exposure’, so the authors need to specify “epigenetic what? and environmental exposure”; suggestions: ‘the epigenetic landscape’, ‘the epigenetic profile’, etc.

As suggested, we added “profile” to epigenetic.

Line 70 – written ‘leukocyte LINE-1 methylation’, again the noun that characterises the methylation is omitted, please add ‘levels’ after methylation to specify what the positive association is with – assuming LINE-1 methylation levels is what the authors are talking about here.

As suggested, we added “levels” after LINE-1 methylation. 

Lines 239-241 – “it has been estimated that LINE-1 insertions account for at least 1 in every 50 humans within a parental germ cell or during early fetal development.“ > the sentence misses something, it does not become clear what LINE-1 insertions account for – mutations or what? Please revise the sentence and change so it makes sense.

We apologize for the confusing sentence that has been rephrased accordingly.

Lines 318-319 – the authors have stated “Although our study suggested LINE-1 methylation as a molecular mechanism underpinning the protective effect of healthy diet” > this is quite a strong statement. There are around 30 million CpGs in the human genome, a large proportion of them are methylated. The genomic stability depends on maintaining the methylation status of many of those CpGs, whether in repeats, promoters, gene bodies or elsewhere. Measuring 3 CpGs in one of the common repeats (of 500,000 copies), which could indeed cause problems when hypomethylated, could definitely suggest one of the mechanisms, but it is a very strong statement to suggest this is the “molecular mechanism underpinning the protective effect of healthy diet”. The authors could make a point by providing some background on numbers (if those 3 CpGs actually occur 500,000 times each and make 1,5 million CpGs in the genome in total this is impressive), but nevertheless, I would tone this down here and also in the abstract. The prudent diet supposes a higher intake of folic acid, which helps to maintain higher global DNA methylation levels (which otherwise tend to erode with aging) – but this effect is most likely seen throughout the whole methylome, not only in LINE-1 (which is also why LINE-1 is used as an indirect measure for the global methylation levels). 

As described in the previous point, we rephrased the conclusion of our manuscript.

This manuscript is a resubmission of an earlier submission. The following is a list of the peer review reports and author responses from that submission.

Round 1

Reviewer 1 Report

The report is a cross-sectional women cohort study aiming to elucidate the relationship between the diet chosen and the consequence of the methylation status of LINE-1. It is of great interest and the conclusion that a positive correlation between intake of “healthy” foods and LINE-1 methylation levels, which in turn was reported negatively in association with chromosomal instability and aberrant genome function.  It will be necessary and important to present additional solid evidence (markers) regarding chromosome and/or genome instability in the leukocytes, then a possible molecular mechanism would be suggested convincingly.

Reviewer 2 Report

In this study the authors study the relationship between dietary intake and patterns with LINE-1 methylation in women from Southern Italy. Here are my comments of this study

1. This study is a correlation study with no causal links.

2. The study cohort is poorly/not at all controlled for environmental and/or behavioural effects.

3. Food Frequency Questionnaire is not an effective/reliable method of data collection.

Reviewer 3 Report

This is an interesting study on the association between diet and LINE-1 DNA methylation and it is reads very well. However, I advise the following revisions:

-        The authors test for normality and then use a mixture of non-parametric tests (eg Kruskal-Wallis and Spearman correlations) and parametric (linear regression). The authors should specify if the data were normal and why they chose those tests. If normality was violated, could they use transformations or other methods? Also, they should use the same method for Figure 3 and table 2 as they are essentially testing the same thing. They should just compare the results of the regression when the covariates are excluded with the model that includes the covariates.

-        All the results of statistical tests need to be reported fully.

-        Include all the variables presented in Table 1 as covariates in the regression. This is because they should be selected based on the association in an independent sample to avoid bias. Also, it could be that for some variables there was not enough power to detect an association so it should not be removed based on a high p-value.

-        Correct the p-values in Figure 2 for multiple comparisons, for example using the Bonferroni threshold or other methods that take into account correlation between the food items.

-        The authors should describe how they chose the CpGs to analyse. In the study by la Rocha they cite and other studies the CpGs in LINE-1 were 8.

-        The authors should add and elaborate on the limitation that as this is an observational study it will still be of a correlative nature. It will be interesting to see the difference between a crude model and the adjusted model that includes all the variables for the variation in both prudent and western diets. It could be that including other factors will diminish the effect size. Even if this is not seen, there could be unmeasured residual confounding that hinders the possibility of making any causal explanation. Education and employment (as well as financial difficulties, although not included here) could still explain the associations found.